# Cartilage and Bone Destruction in Arthritis: Pathogenesis and Treatment Strategy: A Literature Review

**DOI:** 10.3390/cells8080818

**Published:** 2019-08-02

**Authors:** Daisuke Tateiwa, Hideki Yoshikawa, Takashi Kaito

**Affiliations:** Department of Orthopedic Surgery, Osaka University Graduate School of Medicine, 2-2 Yamadaoka, Suita, Osaka 565-0871, Japan

**Keywords:** arthritis, osteoarthritis, rheumatoid arthritis, psoriatic arthritis, osteoclast, cartilage and bone destruction

## Abstract

Arthritis is inflammation of the joints accompanied by osteochondral destruction. It can take many forms, including osteoarthritis, rheumatoid arthritis, and psoriatic arthritis. These diseases share one commonality—osteochondral destruction based on inflammation. The background includes a close interaction between osseous tissues and immune cells through various inflammatory cytokines. However, the tissues and cytokines that play major roles are different in each disease, and as a result, the mechanism of osteochondral destruction also differs. In recent years, there have been many findings regarding not only extracellular signaling pathways but also intracellular signaling pathways. In particular, we anticipate that the intracellular signals of osteoclasts, which play a central role in bone destruction, will become novel therapeutic targets. In this review, we have summarized the pathology of arthritis and the latest findings on the mechanism of osteochondral destruction, as well as present and future therapeutic strategies for these targets.

## 1. Introduction 

Many diseases lead to arthritis, and the most well-known diseases are osteoarthritis (OA), rheumatoid arthritis (RA), and psoriatic arthritis (PsA) [1,2,3]. The morbidity rates of OA and RA are reported to be 10–12% [4,5] and 0.5–1% of the adult population, respectively [6]. The socioeconomic burden of arthritis, as well as the cost of pharmacological and surgical treatment (biological agents, joint arthroplasties, etc.), is enormous [7,8]. Indeed, the cost of OA is estimated to account for 1–2.5% of the gross national product in the United States, Canada, the United Kingdom, France, and Australia [4,9].

These diseases lead to osteochondral destruction via various extracellular and intracellular signals, with inflammation or mechanical stress as the cause. In osteochondral destruction in arthritis, various immune cells and osseous tissues interact closely via mediators, such as tumor necrosis factor (TNF) α, interleukins (ILs), matrix metalloproteinase (MMP), and a disintegrin and metalloproteinase with thrombospondin motifs (ADAMTS) [10]. Treatments that target these mediators play a crucial role in the current pharmacotherapy for RA and PsA.

In recent years, studies have reported evidence of the underlying mechanisms of osteochondral destruction in OA, RA, and PsA. These findings extend not only to extracellular signaling pathways but also to intracellular signaling pathways [11]. Intracellular signaling of osteoclasts, which plays an important role in bone destruction, will be the novel therapeutic targets. Various molecular mechanisms causing cartilage destruction, including mechanical stress and inflammation, have been elucidated. Novel drugs are being developed, and clinical trials are being conducted with such targets.

This review summarizes the latest findings on the pathology of arthritis and the mechanism of osteochondral destruction for OA, RA, and PsA. It also summarizes present and future therapeutic strategies for those targets.

## 2. Osteoarthritis (OA)

### 2.1. Characteristics and Pathology

OA is a disease in which joint cartilage homeostasis cannot be maintained because of joint surface damages, caused by trauma, inflammation, or excessive mechanical stress, including compression forces and tensile strains. It leads to deformation and structural damage [2].

In the past, OA was considered to be a noninflammatory arthropathy, unlike inflammatory diseases, such as RA. However, recent studies have shown that inflammatory cytokines, such as TNF-α and IL-1β, are essential factors in osteochondral destruction in OA [12,13]. Mechanical stress and inflammatory cytokines inhibit the synthesis of collagen and proteoglycan in chondrocytes and promote the production of proteases, such as MMP and aggrecanase [14]. This leads to cartilage destruction [15,16]. OA is now defined as a whole joint disease characterized by cartilage destruction, subchondral bone change, osteophyte formation, and alterations of ligaments and meniscuses [17], but the pathological condition of OA is centered on cartilage destruction.

#### Genetic Factors in OA

The involvement of genetic factors in OA has been reported. A classic twin study described an apparent genetic effect for OA, with a genetic influence ranging from 39–65% after adjusting for known risk factors [18]. To date, genome-wide association studies (GWAS) have identified several genetic variants associated with OA, although it has been observed that the individual risk alleles identified only exerted moderate to slight effects on overall OA susceptibility [19]. This information can be applied to detect individuals at high risk of developing OA, and preventive or interventional treatments subsequently implemented [17].

### 2.2. Cartilage Destruction

Articular cartilages are simple tissues comprising chondrocytes and the extracellular matrix (ECM) without blood vessels. They have a four-layered structure comprising the superficial, middle, deep, and calcified cartilage zones. Chondrocytes are buried in a large amount of ECM and are found sporadically from the superficial zone to the deep zone [17]. When cartilage destruction in OA is observed morphologically, it begins with fibrillation of the superficial zone of the articular cartilages, followed by fissures in the middle and deep zones, and ends with the loss of all layers of cartilage, exposing the subchondral bone to the articular cavity and leading to eburnation [17,20]. Mechanical stress and inflammatory cytokines induce degradation or loss of the ECM, a decline in ECM production, and chondrocyte death [21]. The degradation of the ECM is the most important process in cartilage destruction.

#### 2.2.1. ECM Degradation

The ECM is comprised mainly of collagens and proteoglycans, as well as other less-abundant components, such as elastins, gelatin, and matrix glycoproteins. Type II collagen is the main structural protein of cartilage, forming a network structure with type VI, IX, and XI collagens. Aggrecan and other proteoglycans are tangled within the collagen network structure and draw water to provide compressive resistance [2,22]. Chondrocytes play a central role in regulating cartilage architecture and biochemical composition. The degradation of the ECM (collagens and proteoglycans) is caused by MMP or ADAMTS, which are produced by chondrocytes, synovial fibroblasts, and macrophages [2]. 

#### 2.2.2. ECM Degradation by MMP

MMPs are calcium-dependent zinc-containing endopeptidases, and they degrade the ECM, including collagens and proteoglycans [23]. There are at least 26 types of molecules in the human MMP family, which are all either secreted MMPs or membrane-associated MMPs. These two types are further classified into collagenases, gelatinases, stromelysins, and matrilysins, based on substrate specificity [23]. The expression of MMP-1, MMP-3, MMP-13, and MMP-28 is more heavily promoted in OA cartilage compared to that in normal cartilage [24]. In particular, MMP-13 is involved in strong degradation activity against type II collagen, playing the central role in cartilage destruction [25]. The production of these MMPs is promoted by inflammatory cytokines and growth factors (e.g., TNF-α, IL-1, IL-17, transforming growth factor-β, histamine) [16,24].

#### 2.2.3. ECM Degradation by ADAMTS

The ADAMTS family are secretory metalloproteinases comprising 19 types of molecular species [26]. In cartilage destruction, strong aggrecanases—ADAMTS4 and ADAMTS5—play an important role. Animal experiments comparing ADAMTS4 and ADAMTS5 knockout mice has demonstrated that ADAMTS5 is the stronger aggrecanase [27,28]; however, in humans, ADAMTS4 is highly expressed and is promoted by IL-1 stimulation. Therefore, ADAMTS4 as well as ADAMTS5 is closely involved in OA cartilage destruction in humans [29].

#### 2.2.4. The Role of Toll-like Receptors (TLRs)

TLRs are type I transmembrane receptor proteins that play a key role in the innate immune response by recognizing pathogen-associated molecular patterns (PAMPs). In humans, the TLR family comprises 10 members (TLR1–TLR10), and some of them recognize host-derived damage-associated molecular patterns (DAMPs) that are produced when tissues are damaged [30]. 

Recently, the involvement of TLR4 in the development and progression of OA has been reported [30]. TLR4 was originally identified as a molecule that recognized lipopolysaccharide (LPS), which is a common component of the gram-negative bacterial cell wall. However, a distinct feature of TLR4 is that it binds to several PAMPs other than LPS and to various DAMPs, including high mobility group protein B1 (HMGB1), heat shock protein, free fatty acids, and extracellular matrix degradation products, such as fibronectin and hyaluronic acid [30]. When TLR4 binds to its agonists (PAMPs and DAMPs), it becomes activated. Subsequently, TLR4 signaling activates various transcription factors, including interferon regulatory factor (IRF), mitogen-activated protein kinase (MAPK), and NF-κB, promoting innate immune responses and inducing the production of inflammatory cytokines [30,31]. 

TLR4 is highly expressed in immune cells, but it is also expressed in chondrocytes, synoviocytes, and osteoblasts [32]. It has been shown that TLR4 is highly expressed in cartilage lesions in OA patients [33]. In fact, activation of TLR4 signaling in human chondrocytes promotes the production of inflammatory cytokines and inhibits the synthesis of aggrecan and type II collagen [33]. Furthermore, it has been reported that there is an association between mechanical stress and TLR4 signaling, and that high shear stress promotes TLR4-dependent production of IL-6 in human chondrocytes [32]. 

In cartilage, several DAMPs activate TLR4 signaling, thereby producing inflammatory cytokines, causing cartilage degradation. Small hyaluronic acid fragments can activate TLR4 in chondrocytes, which induces an inflammatory response. However, high molecular weight hyaluronic acid exerts an inhibitory effect on the inflammatory response, suggesting that the action of hyaluronic acid on TLR4 signaling may differ depending on its molecular size [34]. Other TLR4 agonists involved in OA conditions include fibronectin fragments, HMGB1, serum amyloid A, S100-A8, S100-A9, and some plasma proteins contained in synovial fluid [30].

### 2.3. Molecular Mechanisms Associated with OA

Various molecular mechanisms, such as hypoxia-inducible factor-2α (HIF-2α), runt-related transcription factor 2 (Runx2), Hedgehog, parathyroid hormone (PTH), nuclear factor-kappa B (NF-kB), and Notch, are involved in the pathology of OA [35,36,37,38,39,40,41] (Figure 1).

HIF-2α induces the major catabolic mediators of cartilage destruction, such as MMP, ADAMTS, prostaglandin-endoperoxide synthase-2, and nitric oxide synthase [35,36], and is considered to be the central transcriptional activator in cartilage destruction. Biophysical factors, such as mechanical instability induced by destabilization of the medial meniscus surgery, and degradation of the ECM caused by collagenase injection, induce the expression of HIF-2α in osteoarthritic cartilage. Biochemical factors, such as proinflammatory cytokines, also cause Hif-2α expression [35,36]. HIF-2α is expressed in the degenerated cartilage of early-stage OA patients and is garnering attention as the novel therapeutic target for early-stage OA patients [42].

Hypertrophic chondrocytes are often found in the early stage of articular cartilage destruction, from which MMP and ADAMTS are secreted, triggering OA [43]. The essential transcription factor in the hypertrophy of chondrocytes is Runx2. When OA is induced in Runx2 hetero-knockout mice, cartilage destruction is suppressed more than in wild-type mice [37]. A recent report argued that the Hedgehog signal regulates the expression of ADAMTS5 through Runx2 [38]. On the other hand, PTH suppresses the hypertrophy of chondrocytes. When a high dose of PTH is administered to the murine OA model, cartilage destruction is suppressed [39]. These molecular mechanisms involved in the hypertrophy of chondrocytes could become therapeutic targets of OA.

NF-κB is a ubiquitously-expressed transcription factor that regulates inflammation, immune responses, cell proliferation, and survival [44]. The NF-κB family comprises five subunits: RelA/p65, RelB, c-Rel, p50/p105 (NF-κB1), and p52/p100 (NF-κB2). There are two distinct pathways that activate NF-κB, the canonical (classical) pathway and the non-canonical (alternative) pathway. The canonical pathway is activated by inflammatory cytokines and promotes the secretion of various degrading enzymes, including MMP and ADAMTS, while suppressing the expression of ECM-synthesizing molecules, such as Sox9, and the synthesis of type II collagen and aggrecan [45]. In fact, in a murine OA model, knockdown of NF-kBp65 with siRNA suppresses cartilage destruction [40].

Notch is a single-pass transmembrane receptor protein, and Notch family members expressed in chondrocytes of articular cartilage regulate articular cartilage homeostasis [46]. Recent reports disclosed that Notch signaling plays an important role in endochondral ossification, which is a crucial process for OA pathogenesis [41,47]. Notch signaling plays an important role in maintaining the terminal stage of endochondral ossification, including matrix degradation and vascular invasion. When injecting a small compound Notch inhibitor into the murine knee joint, the onset of OA can be prevented [41]. Recently, the Notch-Hes1 signaling pathway has been reported to modulate OA development. Hes1 is a transcription factor that is activated by Notch signaling. Hes1-knockout mice exhibit the suppression of cartilage destruction [47]. Hes1 modulates OA development in cooperation with CaMK2 through induction of inflammatory mediators, including MMP13, ADAMTS5, and IL6. This indicates that the Notch-Hes1 signaling pathway plays a crucial role in OA pathogenesis related to inflammation.

Carminerin (also known as cystatin 10) is a cartilage-specific protein that induces chondrocyte calcification during endochondral ossification [48]. When OA was induced in carminerin-deficient mice, the degradation of cartilage was similar to the level observed in the wild type, but chondrocyte calcification was suppressed during formation of osteoarthritic osteophytes in carminerin-deficient mice [48]. Rather than OA in the limbs, we anticipate that carminerin will become a therapeutic target for spinal OA changes that cause spinal canal stenosis.

### 2.4. Present Therapeutic Strategy for OA

The main pharmacotherapy for OA includes acetaminophen and non-steroidal anti-inflammatory drugs (NSAIDs). When pain cannot be sufficiently controlled with these drugs, serotonin–norepinephrine reuptake inhibitors, tricyclic antidepressants, and opioids may be used [49].

Intra-articular injections of hyaluronic acid and steroids are also used widely [50,51,52]. Hyaluronic acid is a large but simple glycosaminoglycan with high water-holding capacity and viscosity. Hyaluronic acid acts as a lubricant, but its concentrations in OA joints are lower than normal [53]. There are several meta-analyses that have evaluated hyaluronic acid intra-articular injections; however, the results of these studies were not consistent and their efficacy is controversial [50,51].

Surgical management is considered for OA patients who do not attain adequate pain relief and functional improvement with a combination of non-pharmacological and pharmacological treatment [49,54]. Joint replacement surgery is available for patients with severe symptoms or functional limitations associated with poor health-related quality of life despite conservative treatment. Osteotomy and joint preserving surgeries are considered for young and physically active patients with symptomatic OA [54].

### 2.5. Novel OA Therapeutic Strategies

Currently, the main drugs for OA treatment are NSAIDs. However, their long-term use is often difficult because of side effects, such as gastrointestinal bleeding, renal dysfunction, and cardiovascular events [49]. Therefore, the development of novel analgesics that are effective and have fewer side effects is necessary. Alternative therapeutic methods to NSAIDs also need to be established. Here, we have summarized the drugs currently being considered for use as novel OA treatments (summarized in Table 1).

#### 2.5.1. Anti-Inflammatory Cytokines Therapy

Recent studies have demonstrated that inflammation is an important factor in the onset of OA [12,13]. Therefore, treatments that target inflammatory cytokines, such as TNF-α and IL-1, have been examined.

There have been several clinical trials using a human anti-TNF-α monoclonal antibody—adalimumab —on patients with OA [55,56,57]. One study reported that intra-articular adalimumab injections improved clinical symptoms with significantly more efficacy than intra-articular injections of hyaluronic acid in patients with severe knee OA [55]. However, no significant difference was observed in patients with hand OA [56]. Differences in efficacy based on the site of OA, the severity, the method of pharmaceutical administration, and the dose need to be examined.

Diacerein, an IL-1β inhibitor, provided small symptomatic benefit in terms of pain reduction and prevented radiographic joint space narrowing in patients with OA of the hip joint [58]. Other IL-1 inhibitors, such as Anakinra (an IL-1 receptor antagonist) and AMG108 (an IL-1 receptor monoclonal antibody) have been trialed clinically, but favorable results have not been obtained [59,60].

#### 2.5.2. Bisphosphonate

Bisphosphonates have been used for OA to suppress the activity of osteoclasts, delay bone remodeling, and provide chondroprotective effects. Zoledronic acid reportedly improves pain and reduces bone marrow lesions in patients with knee OA [61]. In recent studies, the efficacy of clodronate, a first-generation non-nitrogenous bisphosphonate, has been investigated. In randomized controlled trials on patients with knee [62] and hand OA [63], improvements in pain and joint function were reported. Recent reports disclosed that clodronate inhibits vesicular adenosine triphosphate release from neurons, microglia, and immune cells and exerts therapeutic effects on neuropathic pain, inflammatory pain, and chronic inflammation [64]. Furthermore, clodronate promotes the expression of Sox9, which is essential for chondrocyte differentiation, and leads to chondroprotection [65].

#### 2.5.3. Strontium Ranelate

Strontium ranelate, a strontium (II) salt of ranelic acid, is known as an antiosteoporotic agent. Strontium renalate is unique in that it stimulates osteoblastic bone formation and simultaneously reduces osteoclastic bone resorption [66]. Furthermore, strontium ranelate induces cartilage formation through an ionic effect [67]. In a randomized controlled trial on patients with knee OA, strontium ranelate therapy for a three-year period prevented radiographic joint space narrowing and improved clinical symptoms [68]. However, the increased risk of cardiac events, such as venous thrombosis, pulmonary embolism, and myocardial infarction, has been reported in randomized controlled trials, but not in real life; therefore, further studies are warranted [69].

#### 2.5.4. Anti-Nerve Growth Factor (NGF) Antibody

NGF is a target-derived factor involved in the survival and maintenance of peripheral neurons. The expression of NGF is markedly increased in several pain conditions, including OA, and it sensitizes the nociceptive system [70]. In the OA condition, chondrocytes and synovial cells produce NGF through inflammatory factors and mechanical stress [71], and NGF promotes nerve growth in intra-articular tissues, enhancing pain sensitivity [72]. Therefore, NGF has been investigated as a novel pain target for OA. The effects of two anti-NGF antibodies—tanezumab and fasinumab—on OA have been examined in several clinical trials, and improvements in joint pain and function have been reported. [73,74,75]. Tanezumab and fasinumab were generally well tolerated. The most common adverse events in tanezumab were headache, upper respiratory tract infections, and sensory abnormalities [73]. We anticipate that anti-NGF therapy will be a novel therapeutic strategy that targets not only osteochondral destruction, but also OA pain.

#### 2.5.5. Therapies Targeting TLR Signaling

TLR signaling, in particular TLR4-mediated signaling, is involved in the development and progression of OA. Recently, the efficacy of treatments targeting TLR4 signaling has been reported in preclinical studies [76,77,78]. Pep-1 (hyaluronic acid-binding peptide) inhibits the degradation of hyaluronic acid and consequently reduces the small hyaluronic acid fragmentation ratio [76]. Peroxisome proliferator-activated receptor γ agonists (genistein and rosiglitazone) inhibit TLR4 signaling by reducing serum amyloid A expression [77]. Vasoactive intestinal polypeptide (VIP) decreases TNF-α-induced TLR4 expression in synoviocytes from OA patients [78]. Although the efficacy of new compounds that regulate TLR4 signaling, such as oleochanthal, 6-shogaol, boswellic acid, quercetin, and kampferol, has been reported, the mechanisms have not yet been fully elucidated [30,79,80].

#### 2.5.6. miRNA

miRNAs are small non-coding single-stranded RNAs that typically contain 18–25 nucleotides. They regulate target gene expression specifically at the post-transcriptional level [81]. In humans, more than 2000 types of miRNA have been identified [82]. One miRNA targets multiple mRNAs, indicating that there are complex gene regulatory networks via miRNA [81]. In recent years, miRNA has gained attention as an important molecule associated with the onset and progression of OA [83,84]. miR-140 is a miRNA involved in maintaining homeostasis of the cartilage [85]. miR-140 protects cartilage by suppressing the expression of ADAMTS5; however, the expression of miR-140 is suppressed in the cartilage of patients with OA [85]. In miR-140 knockout mice, cartilage destruction progresses, whereas cartilage destruction is suppressed in transgenic mice with excessive expression of miR-140. In addition, other miRNAs involved in homeostasis and the structure of cartilage in OA, such as miR-16, miR-23b, miR-26a, miR-223, miR-377, miR-483, and miR-509, have been the subject of many reports and are expected to be possible novel therapeutic targets for OA [86]. 

Besides the functional analyses of miRNA that are actively performed, the development of the drug delivery system (DDS) is essential. The reason why practical use of nucleic acid drugs, including miRNA, is difficult is that they are easily broken down by nuclease in the blood and cannot effectively reach local sites. In recent years, there have been reports on the use of exosome as a DDS for mRNA and miRNA [87,88]. Exosomes are small membrane vesicles about 30–100 nm in size that are secreted by a variety of cells and carry various proteins, mRNAs, and miRNAs to remote cells like a “message in a bottle” [88]. Exosome is an ideal carrier that can protect miRNAs from ribonuclease in the bodily fluid. Furthermore, it would be possible to inject nucleic acid drugs directly into the articular cavity since joints are closed environments surrounded by the articular capsule [89]. Novel therapeutic methods that combine these approaches are anticipated.

#### 2.5.7. Cell-Based Therapies

Cell-based therapies for OA have advanced remarkably in recent years. Currently, the only cell-based therapy approved by the Food and Drug Administration (FDA) is autologous chondrocyte implantation (ACI), whereby cultured chondrocytes are applied to the injured cartilage area [90]. Although long-term results are lacking, several clinical trials have reported the short-term efficacy of ACI for OA treatment [91].

In recent years, mesenchymal stem cells (MSCs) have attracted the most attention in cell-based therapies for OA [92,93]. MSCs are well known as multipotent cells with self-replication potential. Furthermore, MSCs possess immunomodulatory properties that suppress the immune response and reduce the production of inflammatory cytokines. Although MSCs can be isolated from various tissues, bone marrow-derived MSCs (BM-MSCs) and adipose-derived MSCs (AD-MSCs) are most commonly used [94]. In a comparison between BM-MSCs and AD-MSCs, BM-MSCs showed superior chondrogenic and osteogenic ability, while AD-MSCs were superior in terms of immunomodulatory properties [95].

There are two main methods to deliver MSCs into the joints: intra-articular injection of MSCs in suspension and surgical procedures that implant MSC-laden scaffolds [93]. Intra-articular injection of MSCs is relatively simple, less invasive, and more cost effective compared with surgical procedures [93]. On the contrary, in surgical procedures using scaffolds, the properties of the scaffolds can result in excellent therapeutic effects. An ideal scaffold should initially promote cell migration and provide biological, chemical, and mechanical support to MSCs. It should also be finally absorbed or biodegraded so as not to hamper new cartilage formation or induce an inflammatory response [96].

According to recent preclinical and clinical trials, OA treatment using MSCs has demonstrated a significant improvement in clinical results, with minimal adverse events [92,93,97,98]. However, more reliable studies with randomized controls and larger sample sizes are required for robust evidence.

## 3. Rheumatoid Arthritis

### 3.1. Characteristics and Pathology

RA is one of the most widespread inflammatory autoimmune diseases, and its major symptoms are systemic joint swelling, pain, and disability. RA is characterized by proliferative synovitis causing cartilage and bone destruction. During the active phase of RA, pannus forms at the site where synovial cells have proliferated [1]. The inflamed synovium comprises T-cells, synovial fibroblasts, and macrophages that produce inflammatory cytokines, such as TNF-α, IL-1, IL-6, IL-17, and macrophage colony-stimulating factor (M-CSF). These inflammatory cytokines activate osteoclasts, leading to bone destruction [1] (Figure 2).

In RA, there is an excessive immune response of T-cells. CD4+ T-cells consist of T-helper (Th) cells that promote the immune responses and regulatory T-cells (Treg) that regulate these immune responses [99]. Th cells have subsets such as Th1, Th2, and Th17 cells. Th17 cells play an important functional role in phlogistic action. Naive T-cells differentiate into Th17 cells through IL-1β, IL-6, IL-21, and transforming growth factor-β (TGF-β). IL-17 produced by Th17 activates inflammation by acting on various immune cells and activates osteoclasts by inducing the receptor activator of nuclear factor kappa B ligand (RANKL) in synovial fibroblasts [100]. In addition, it has been recently reported that exFoxp3Th17 promotes osteoclast differentiation three times more effectively than Th17 [101]. On the other hand, cytokines, such as IFN-γ, IL-4, and cytotoxic T-lymphocyte-associated protein 4 (CTLA-4), produced by Th1, Th2, and Treg, respectively, regulate osteoclast differentiation. In RA, there is an imbalance in Th17/Treg, where Th17 is activated markedly more than Treg [10].

#### Genetic Factors in RA

Genetic factors are closely involved in the onset of RA. It is estimated that genetic factors contribute to 50% of the risk of developing RA [102]. A nationwide twin study reported that disease concordance in monozygotic twins (15.4%) is considerably higher than in dizygotic twins (3.6%) [103]. Among MHC class II genes, HLA-DR1 and HLA-DR4 are most closely connected to RA susceptibility [104]. To date, in multiple populations, the GWAS have identified more than 35 RA susceptibility loci, such as *HLA-DRB1*, *STAT4*, *PTPN22*, *PAD14*, and *CCR6* [105,106].

### 3.2. Mechanism of Bone and Cartilage Destruction

Osteoclasts play a critical role in bone destruction in RA. Osteoclasts, which are multinucleated giant cells, are derived from hematopoietic stem cells and develop from the same monocyte lineage progenitor cells [107]. Examining the sites of bone erosion at the bone-pannus interface in patients with RA, tartrate resistant acid phosphatase (TRAP)-positive, calcitonin receptor-positive, and Cathepsin K-positive multinucleated giant cells, that is, osteoclasts, have been observed [108]. These findings demonstrate that bone destruction in RA is caused not by direct invasion by synovium but by the effects of osteoclasts [108].

For osteoclast differentiation, RANKL, which is known as a member of the TNF family, is essential, along with M-CSF [109]. Inflammatory cytokines, such as TNF-α, IL-1, IL-6, and IL-17 induce excessive RANKL on the membrane of synovial fibroblasts or osteoblasts. Direct cell-to-cell contact between osteoblasts and osteoclast progenitors activates the receptor activator of nuclear factor kappa B (RANK)/RANKL pathway and consequently promotes osteoclast differentiation [109,110]. The importance of the RANK/RANKL pathway in the living body has been demonstrated by a murine genetic study [111]. In RANKL and RANK knockout mice, osteoclasts were missing and osteopetrosis was identified [111,112]. When osteoprotegerin, which is a decoy receptor of RANKL, was administered to arthritis rat models, bone destruction was regulated [112]. These findings show that the RANK/RANKL pathway plays a critical role in osteoclast differentiation.

M-CSF, which is produced by osteoblasts/stromal cells, is also critical for osteoclastogenesis, as demonstrated by analyses of osteopetrotic (op/op) mice lacking functionally-active M-CSF [113]. M-CSF is indispensable not only for proliferation of osteoclast progenitor cells, but also for their differentiation into mature osteoclasts [114].

However, arthritis and bone destruction need to be differentiated. In one study, cFos-deficient mice with no osteoclasts were mated with TNF-α transgenic mice—model mice for arthritis—and despite the onset of arthritis, bone destruction was notably regulated [115]. In other words, even if residual inflammation exists, joint destruction might be prevented by suppressing the activity of osteoclasts.

Cartilage destruction is caused by MMP or ADAMTS, which are produced by chondrocytes, synovial fibroblasts, and synovial macrophages. Epigenetic changes that maintain high levels of MMP expression have been found in RA synovial fibroblasts [116]. 

### 3.3. Intracellular Signals of Osteoclasts

Intracellular signals of osteoclasts may be the novel therapeutic target of RA (Figure 3). First, when RANK is stimulated, the TNF receptor-associated factor (TRAF) family is recruited to RANK [117]. TRAF6 is especially important for RANK/RANKL signaling, as demonstrated by a study in which TRAF6-deficient mice developed osteopetrosis [118]. When TRAF6 is recruited to RANK, downstream nuclear factor-kappa B (NF-κB), mitogen-activated kinases (MAPKs), and activator protein1 (AP-1) are activated. This ultimately activates the nuclear factor of activated T-cell c1 (NFATc1), which is the master transcription factor of osteoclast differentiation [119,120]. Activated NFATc1 couples with other transcription factors, such as cyclic adenosine monophosphate-response element-binding protein (CREB), AP-1, microphthalmia-associated transcription factor (MITF), and PU.1, and promotes the expression of osteoclast-specific genes, such as TRAP, Cathepsin K, the calcitonin receptor, and osteoclast-associated receptor (OSCAR) [121].

In addition to M-CSF and RANKL, the immunoreceptor tyrosine-based activation motif (ITAM) that activates calcium signaling is important for osteoclast differentiation [122]. Adaptor molecules with ITAM, such as Fc receptor common gamma chain (FcRγ) and DNAX-activating protein of 12 (DAP12), are associated with immunoglobulin-like receptors, such as OSCAR, paired immunoglobulin-like receptor-A (PIR-A), triggering receptor expressed on myeloid cells 2 (TREM2), and signal regulatory protein beta 1 (SIRPβ1), subsequently activating spleen tyrosine kinase (Syk). This is followed by Bruton’s tyrosine kinase (Btk) and tyrosine kinase expressed in hepatocellular carcinoma (Tec) integrating RANK/RANKL signaling with ITAM signaling, and activation of calcium signaling [123]. When the activation of calcium signaling is maintained, calcineurin is activated. Activated calcineurin induces nuclear translocation of dephosphorylated NFATc1, and ultimately dephosphorylated NFATc1 allows for the sustained autoamplification of NFATc1. ITAM signals cannot induce osteoclastogenesis by themselves, but these are essential for osteoclastogenesis for regulating the threshold of RANK-initiated calcium signals [11].

### 3.4. The Current Therapeutic Strategy for RA

The “anchor drug” of RA therapy was methotrexate (MTX), but RA therapy has drastically changed in recent years with the arrival of biological agents [124,125]. MTX, which was first used for RA treatment more than 50 years ago, is considered to be the most important of the conventional synthetic disease-modifying antirheumatic drugs (csDMARDs) and one of the pivotal drugs in the present RA therapeutic strategy. Considering the cost-effectiveness and adverse effects of biological disease modifying anti rheumatic drug (bDMARD) therapy, the use of MTX is recommended as the first line therapy for RA, even now [126].

However, if MTX is ineffective because of an inadequate response, contraindications, intolerance, or adverse effects, bDMARDs or targeted synthetic disease modifying anti rheumatic drugs (tsDMARDs) should be considered [126]. The recent development of bDMARDs and tsDMARDs is remarkable and has brought about a wide range of options for RA treatment. Here, we summarize the use of bDMARDs and tsDMARDs in present RA treatment.

#### 3.4.1. TNF inhibitors

TNF inhibitors, which were developed as an initial bDMARD, brought about a drastic paradigm shift in the treatment of RA at the end of the last century [127]. There are five major TNF-α inhibitors: infliximab, etanercept, adalimumab, golimumab, and certolizumab pegol. Infliximab, adalimumab, golimumab, and certolizumab pegol are anti-TNF-α monoclonal antibodies [127,128]. They neutralize the biological activities of TNF-α by binding with soluble TNF-α. Infliximab, adalimumab, and golimumab also bind to membrane-bound TNF-α and induce apoptosis, antibody-dependent cellular cytotoxicity, and complement-dependent cytotoxicity in TNF-α-producing cells. Etanercept is a soluble receptor, which functions as a decoy receptor of TNF-α [127,129]. 

Despite the fact that TNF inhibitors are contraindicated in patients with heart failure and increase the risk of severe adverse events, such as infection [130,131], they are widely used as the standard care, especially when MTX treatment has failed. Currently, about 70–80% of RA patients receive the combination therapy of TNF inhibitors and MTX [127,128]. 

#### 3.4.2. IL-6 Inhibitors

Toll-like receptor stimulation with pro-inflammatory cytokines, such as IL-1, IL-6, and TNF-α, induces the production of IL-6 in lymphocytes, macrophages, and synovial cells. IL-6 has diverse functions, such as acute-phase response, activation of immune reaction, and hematopoiesis [132,133]. IL-6 signaling is triggered when IL-6 binds specifically to the IL-6 receptors. There are two types of IL-6 receptor—the transmembrane IL-6 receptor and the soluble IL-6 receptor. After IL-6 binds to either receptor, the resultant complex induces activation of gp130, and ultimately the JAK/signal transducer and the activator of transcription 3 (STAT3) pathway are activated downstream [134]. 

IL-6 regulates the initiation of acute-phase responses. IL-6 signaling induces the production of acute-phase proteins, such as C-reactive protein and fibrinogen, which are known as biological markers of inflammation [135]. As for the effect of IL-6 on the immune response, IL-6 induces the differentiation of naive T-cells into Th17 cells [136]. The imbalance in Th17/Treg, where Th17 is activated significantly more than Treg, is pathologically involved in the development of RA [99].

Humanized anti-IL-6 receptor monoclonal antibodies, such as tocilizumab and sarilumab, bind to both transmembrane IL-6 receptors and soluble IL-6 receptors to block IL-6 mediated signal transduction [137]. According to recent clinical trials, sarilumab is effective for RA patients with an inadequate response not only to MTX, but also to TNF inhibitors [138].

#### 3.4.3. Janus Kinase (JAK) Inhibitors

The JAK/STAT pathway is the major signaling cascade for various cytokines and growth factors. JAK is a receptor tyrosine kinase that mediates intracellular signals via a transcription factor, STAT [139]. The JAK family comprises JAK1, JAK2, JAK3, and tyrosine kinase 2 (Tyk2), where JAK1, JAK2, and Tyk2 are expressed ubiquitously, whereas JAK3 is only expressed within hematopoietic cells [140].

Tofacitinib and baricitinib, which are presently used to treat RA, are potent oral JAK inhibitors. While tofacitinib inhibits JAK1, JAK2, JAK3, and Tyk2, baricitinib selectively inhibits JAK1 and JAK2 [141]. According to several clinical trials, baricitinib has shown to improve arthritis symptoms not only in DMARD-naive patients, but also in patients with an inadequate response to MTX, csDMARDs, or TNF inhibitors [142,143,144]. Furthermore, baricitinib has shown significantly greater improvements in patients with an inadequate response to MTX compared to adalimumab [145].

#### 3.4.4. T-cell Activation Inhibitors 

Abatacept is a fully soluble human fusion protein in which the Fc region of IgG is fused to the extracellular domain of CTLA4. Co-stimulatory signals are essential for T-cell activation, and without co-stimulatory signals, T-cells undergo apoptosis. CD28, which is expressed in T-cells, is the archetypal co-stimulatory molecule that binds to CD80 and CD86 on the surface of antigen-presenting cells. CTLA4 contained in abatacept binds to CD80 and CD86 with higher affinity than CD28, thereby acting as a negative regulator of the CD80/CD86:CD28 co-stimulatory signal [146]. While TNF-α, IL-6, and JAK inhibitors directly regulate the generation and bioactivity of cytokines, abatacept indirectly contributes to immunosuppression by regulating the activity of T-cells that induce production of cytokines, autoantibodies, and inflammatory proteins.

In MTX refractory patients with RA, treatment with abatacept was well-tolerated over a five year period and provided significantly greater improvements in RA symptoms and disease activity [147]. A recent study examined the characteristics of T-cells targeted by abatacept and the predictors of therapeutic response to abatacept [148]. Abatacept targets CD4+ CD28+ T follicular helper-like cells, and on the other hand, CD4+ CD28- cells represent the resistance against abatacept. The difference in the characteristics of T-cells can be a potential predictor of abatacept resistance [148].

### 3.5. Novel Therapeutic Approches for RA

The present therapeutic agent for RA mainly targets extracellular inflammatory cytokines, such as TNF-α, IL-1, IL-6, IL-12, IL-17, and IL-23. In recent years, the intracellular signals of various cells involved in the pathology of RA have been elucidated [11]. Novel therapies that target these signals are garnering attention. Here, we have summarized new therapies that mainly target intracellular signals in osteoclasts and play a key role in bone destruction in RA, as well as immune cells that induce inflammation (summarized in Table 1).

#### 3.5.1. Btk Inhibitors

Btk belongs to the Tec kinase family and is an intracellular enzyme that is expressed mainly in hematopoietic cells, including B-cells. Via B-cell receptors, it plays an important role in cytokine production and expression of co-stimulators [149]. It is also expressed in myeloid cells, such as monocytes, macrophages, neutrophils, and mast cells [150]. As Btk/Tec is important in the differentiation and activation of osteoclasts in terms of integrating RANK/RANKL signaling with ITAM signaling, it is considered one of the potential targets for RA treatment [151]. Currently, Btk inhibitors, such as poseltinib (HM71224), are in clinical development [152].

#### 3.5.2. Syk Inhibitors

Syk is a non-receptor tyrosine kinase belonging to the Src family, which is involved in various signal transductions via the cell surface receptors, such as B-cell receptor and FcRγ [153]. Syk has a high affinity to ITAM and is involved in signal transduction for B-cell receptors, FcRγ, DAP12, and integrin with ITAM [154]. Although the efficacy of Syk inhibitors, such as fostamatinib disodium (R406), on RA has been reported [155], it has not been used due to its side effects, such as infections, hypertension, diarrhea, nausea, and headache [156].

#### 3.5.3. Phosphoinositide 3-Kinase (PI3K) Inhibitors

PI3K, an enzyme that phosphorylates phosphoinositides, plays an important role in regulating cellular activation, proliferation, and migration in various types of cells, including immunocompetent cells [157]. Pathogenic evidence of the involvement of PI3K/Akt signaling pathways has been reported in patients with RA. PIP3, which is generated by PI3K, activates Akt. High-level expression of activated Akt in the synovium of RA patients indicates that it is involved in synovium proliferation and inflammatory cell infiltration [158]. The efficacy of PI3K inhibitors, such as ZSTK474, has been demonstrated in mice with collagen-induced arthritis [159]. In vitro, PI3K inhibitors suppressed osteoclast formation and proliferation of B lymphocytes and synovial fibroblasts [159]. These findings indicate that PI3K inhibitors might be a novel therapeutic agent against RA. Clinical trials of PI3K inhibitors for RA have not yet been conducted.

#### 3.5.4. MicroRNA (miRNA)

In the near future, we anticipate that miRNAs will become the novel therapeutic targets of RA. Various miRNAs associated with inflammatory cytokines, synovial cell proliferation, and osteoclast differentiation have been reported, and their application in RA treatment has been attempted [160]. miR-146 and miR-155 are quoted as representative miRNAs related to the RA condition [161,162]. miR-146 expression is induced by TNF-α stimulation via NF-kB, which functions as negative feedback that stops the inflammatory stimulation by targeting TRAF6 [161]. The high-level expression of miR-155 has been reported in the synovial membrane tissues in RA patients [163]. Overexpression of miR-155 in RA synovial monocytes and macrophages enhances the production of TNF-α, IL-6, IL-1β, and IL-8. On the other hand, inhibition of miR-155 suppresses the production of TNF-α [164]. Src homology-2 domain-containing inositol 5-phosphatase 1 (SHIP1) and suppressor of cytokine signaling 1 (SOCS1) have been identified as major targets of miR-155 [165,166].

#### 3.5.5. Histone Deacetylase (HDAC) Inhibitors

For the transcription reaction of genes to occur, the chromatin structure needs to relax, and the transcription factor must come in contact with DNA. HDAC regulates the relaxation of the chromatin structure [167]. HDAC inhibitors, such as trichostatin A and givinostat (ITF2357), promote the degradation of mRNA and regulate the generation of inflammatory cytokines in RA synovial fibroblasts and macrophages [168]. Clinical trials of HDAC inhibitors of RA have not been conducted, but the efficacy and safety of givinostat have been reported in clinical trials for juvenile idiopathic arthritis [169]. In the trial, givinostat was safe and well tolerated. The most common adverse events were short-lived, self-limited respiratory or gastrointestinal disturbances. Currently, HDAC3 and HDAC6 inhibitors are anticipated to become novel RA therapeutic agents [170].

## 4. Psoriatic Arthritis (PsA)

### 4.1. Characteristics and Pathology

PsA is a type of spondyloarthritis (SpA), similar to ankylosing spondylitis, reactive arthritis, and enteropathic arthritis [3]. PsA is a chronic autoimmune inflammatory disease associated with psoriatic dermal symptoms, arthritis, and enthesitis. There is progressive, irreversible bone destruction, which leads to joint deformation and dysfunction [171]. The clinical symptoms of PsA are diverse, including arthritis, enthesitis, dactylitis, osteitis, uveitis, and nail dystrophy. Synovitis is the main feature of the RA condition, but the main feature of PsA is enthesitis, which is inflammation of the connective tissue between the bone and tendon [3]. While bone erosion is the characteristic bony change in RA, in PsA, it is not limited to bone erosion but is characterized by subsequent novel bone growth within the same environment [172].

In the pathology of PsA, Th17 cells and the associated IL-23/IL-17 axis are important [3,173,174] (Figure 4). Naive T-cells differentiate into Th17 cells through IL-1β, IL-6, IL-21, and TGF-β, and IL-23 is necessary for survival and proliferation of Th17 cells [175,176,177]. Th17 cells then produce various inflammatory cytokines, such as IL-17A, IL-17F, IL-21, IL-22, IL-25, IL-26, and TGF-β [178,179]. IL-17 drives synovial fibroblasts and macrophages to promote further production of inflammatory cytokines, such as IL1-β, IL-6, and TNF-α, and it ultimately causes osteochondral destruction by inducing the expression of MMPs, ADAMTS, and RANKL [10,180]. Th17 cells are enriched in the circulation and synovial fluid of PsA patients. Given the evidence above, it is clear that Th17 cells play a crucial role in the pathogenesis of PsA [179].

#### Genetic and Environmental Factors in PsA

Genetic factors are closely involved in the onset of PsA. In identical twins, the concordance rate is estimated to be 80–100% [181,182]. The most closely associated disease-susceptibility gene is HLA-Cw6, but there are also correlations with MHC class I genes, HLA-B27, and HLA-B39 [183,184]. Environmental factors involved are trauma, infections, obesity, and smoking [185]. Considering that trauma is an independent risk factor, it is quite likely that physical stimulation is involved in the onset of enthesitis [186].

### 4.2. Enthesitis and IL-23

IL-23 is closely involved in the localized inflammation of the tendon-bone attachments (entheses) and plays an essential role in enthesitis [3]. IL-23 acts on entheseal resident T-cells, which are expressing the IL-23 receptor, RAR-related orphan receptor γt, and stem cell antigen 1. Then activated entheseal resident T-cells (ROR-γt+CD3+CD4−CD8 T-cells) produce inflammatory cytokines, such as IL-6, IL-17, and IL-22, and chemokines, such as C-X-C motif ligand 1, to induce inflammation in entheses [187]. Even without other cytokines, ROR-γt+CD3+CD4−CD8 T-cells become activated by IL-23 stimulation alone, indicating that IL-23 is closely involved in the onset of enthesitis [187,188].

### 4.3. Bone Destruction in PsA

The synovium of patients with PsA has high expression of IL-17A and IL-17 receptors, and IL-17A is closely involved in osteochondral destruction in PsA [189,190]. When IL-17A is over-expressed in mice, they present with psoriatic features, such as epidermal hyperplasia, accompanied by parakeratosis and Munro’s micro-abscesses formation. At the same time, the proliferation of osteoclast precursors is induced, leading to bone destruction. These skin and bone pathologies are typically observed in PsA [191]. On the other hand, when arthritis is induced in IL-17A deficient mice, they show more periosteal bone formation than wild-type mice [192]. IL-17A inhibits osteoblast differentiation in vitro, inducing the expression of secreted Frizzled-Related Protein 1, which is an inhibitor of the Wnt/β-catenin pathway and contributes to inhibition of osteoblast differentiation [192]. IL-17A also activates osteoclasts by inducing the RANKL expression in osteoblasts and synovial fibroblasts, causing bone destruction [10,180].

### 4.4. Bone Formation in PsA

In PsA, bone erosion is followed by new bone formation. This phenomenon makes PsA definitively different to RA. It is suggested that IL-22 is closely involved in bone formation in entheses or around articular cartilages. Enthesis-resident ROR-γt+CD3+CD4−CD8 T-cells not only produce IL-17, but also IL-22 by IL-23 stimulation [187]. IL-22 promotes the proliferation of human mesenchymal stem cells and induces differentiation to osteoblasts [193]. Furthermore, it promotes the ossification of enthesis and periosteum via the signal transducer and activator of transcription-3 (STAT3) of osteoblasts [187].

Involvement of bone morphogenetic protein (BMP) has also been reported [194,195]. BMP, a member of the TGF-β superfamily, is known as a key regulator of bone formation, strongly inducing the expression of osteogenic transcription factors, such as Runx2 and osterix. Endochondral ossification is observed in bone formation sites of SpA patients, and BMP-2, BMP-6, and TGF-β are highly expressed there [194,195]. When IL-1β or TNF-α acts on synovial fibroblasts derived from patients with SpA, the expression of BMP-2 or BMP-6 is induced [195]. In animal experiments, Noggin, which is a non-specific BMP antagonist, suppressed bone formation in sites of ankylosing enthesis [196].

The mechanism of bone formation in PsA has not been sufficiently elucidated; however, because it causes articular dysfunction, it is an important concern. Bone formation in entheses makes PsA definitively different from RA, and the mechanism can be a novel therapeutic target.

### 4.5. Present Therapeutic Strategy for PsA

The present therapeutic agents for PsA include NSAIDs, csDMARDs (such as MTX and salazosulfapyridine), phosphodiesterase 4 (PDE4) inhibitors, and biological agents. NSAIDs are initially used, and if the efficacy is insufficient, csDMARDs are subsequently used [197]. PDE4 inhibitors are generally used for psoriasis or a mild case of PsA. If the effects of csDMARDs are insufficient, biological agents are chosen. However, the order of administration of therapeutic agents needs to be considered with flexibility depending on the target pathology [197,198].

#### 4.5.1. TNF-α Inhibitors

Similar to RA, TNF-α inhibitors have brought about great improvements in the treatment of PsA, and their efficacy and safety have been confirmed [197,198]. However, it is difficult to say that TNF-α inhibitors provide long-term efficacy in most PsA patients. According to the Danish Nationwide DANBIO registry, 39% of PsA patients switched from TNF-α inhibitors to other biological agents during a median follow-up of 2.3 years, and the median drug survival of the first TNF-α inhibitors was 2.2 years [199]. Furthermore, after switching to alternative biological agents, response rates were lower and drug survival times shorter [199]. Not all PsA patients show a satisfactory response to TNF inhibitors. Therefore, we anticipate novel drugs that target other molecular pathways to be developed in the future. 

#### 4.5.2. Anti-IL-23/IL-17 Therapy

Recently, it was discovered that the IL-23/IL-17 axis is closely involved in the pathology of PsA, and the efficacy of treatments targeting the IL-23/IL-17 axis has been reported [3,173,174].

IL-23 is considered to be closely involved in enthesitis in PsA pathology, and there are two representative drugs targeting IL-23—ustekinumab and guselkmab [200,201,202]. Ustekinumab, a fully humanized monoclonal antibody targeting the p40 subunit of IL-12 and IL-23, prevents both IL-12 and IL-23 from binding to the receptors and suppresses the activation of the Th1 and Th17 inflammation pathways [200,201]. Guselkmab, a fully human immunoglobulin G1 λ monoclonal antibody directed against the p19 subunit of IL-23, inhibits the mediated IL-23 signaling pathway more selectively [202].

IL-17 is supposed to have a crucial role in PsA pathology, especially in bone destruction, and there are several types of drugs targeting the IL-17-related signaling pathway. Secukinumab and ixekizumab are human monoclonal antibodies targeting IL-17A [203,204]. Bimekizumab is a human monoclonal antibody targeting both IL-17A and IL-17F, and dual inhibition of both isoforms is expected to be more effective than inhibition of either of them in isolation [205]. Broadalumab is a human monoclonal antibody that targets the IL-17 receptor, therefore it potentially inhibits IL-17A, IL-17F, and IL-17E [206]. 

All these drugs targeting IL-23 or IL-17 show good therapeutic results in clinical trials and are the main agents of the present PsA treatment, along with TNF-α inhibitors [200,201,202,203,204,205,206]. These drugs targeting the IL-23/IL-17 axis are highly effective for treating PsA; however, their efficacy for treating RA is limited [207,208,209]. This implies that PsA and RA may be similar, but different at the same time.

### 4.6. Novel Therapeutic Targets in PsA

#### 4.6.1. Using Appropriate Biological Agents

There are multiple biological agents for PsA that show similarly high levels of therapeutic effects [200,201,202,203,204,205,206]. However, the appropriate selection of these drugs remains a major challenge [197,198]. Currently, a double-blind randomized controlled trial that compares the efficacy of adalimumab (a novel TNF-α inhibitor) and secukinumab (IL-17A inhibitor) in active PsA patients who are naive to biological therapies is in progress (EXCEED1) [210]. New evidence regarding the appropriate use of adalimumab and secukinumab is much anticipated.

One recent study reported the efficacy of precision medicine that selects biological drugs based on the characteristic lymphocyte phenotype of patients with PsA [211]. In this study, PsA patients were classified into four groups according to peripheral blood T-cell phenotypes, and different biological agents were selected corresponding to each group. This strategic treatment was more effective than standard biological agent therapy. If we could efficiently choose the optimum treatment for each patient, it would not only reduce the burden on patients, but also reduce medical costs.

#### 4.6.2. Novel Therapeutic Agents

As for novel therapeutic agents of PsA, clinical trials are being conducted on JAK inhibitors, IL-6 inhibitors, and abatacept, which are already being used to treat RA [212,213,214,215] (summarized in Table 1).

Two recent studies have reported that tofacitinib, an oral JAK inhibitor, showed efficacy not only in TNF inhibitor naive patients, but also in patients with inadequate response to TNF inhibitors [213,214]. Clazakizumab, a monoclonal antibody that has high affinity and specificity for IL-6, has displayed efficacy, especially in musculoskeletal features [212]. Abatacept, a selective T-cell activation inhibitor, provided a beneficial effect on musculoskeletal manifestations and was well-tolerated in both patients naive to TNF inhibitors and those experienced with TNF inhibitors [215].

In addition, novel PsA treatments are being developed, such as Tyk2 inhibitors, chemokine (CC motif) ligand 20 (CCL20) inhibitors, and fecal microbiota transplantation [216,217,218,219,220] (summarized in Table 1).

Tyk2, an intracellular signaling enzyme, mediates signaling downstream of the receptors for IL12, IL-23, and type I and III interferons [221]. BMS-986165, a selective oral inhibitor of Tyk2, had a good therapeutic result in psoriasis that was the same as the results for other biological agents, such as ustekinumab and adalimumab, although an increased risk of mild-to-moderate acne has been reported in the treatment groups [222]. A clinical trial evaluating the efficacy and safety of BMS-986165 in patients with active PsA is now in progress [217]. PF-06700841, a potent selective inhibitor of Tyk2/JAK1, is also expected to be a novel therapeutic agent [218].

CCL20, which is a small cytokine, is a member of the CC chemokine family. CCL20 shows chemotaxis and attracts immune cells, such as T-cells, B-cells, natural killer cells, and dendritic cells, to inflamed tissue [223]. CCL20 is highly expressed in inflamed tissue in PsA. GSK3050002, a humanized immunoglobulin G monoclonal antibody, binds to CCL20 and inhibits the movement of inflammatory cells into inflamed tissues and is expected to be a novel therapeutic agent against PsA [219]. 

Recently, the complex role of the microbiota in the immune system has been reported [224], and abnormal intestinal microbiota may cause the activation of the inflammatory pathways in PsA. Indeed, in PsA patients, the presence of intestinal inflammation has been reported. Furthermore, a recent study has reported that Akkermansia muciniphila, which is an intestinal bacterium playing an important role in maintaining gut homeostasis, was significantly reduced in patients with psoriasis [225]. A study evaluating the efficacy and safety of fecal microbiota transplantation in PsA patients is currently in progress [220].

## 5. Conclusions

Recent studies continue to further elucidate the mechanisms of osteochondral destruction in arthritis. The ultimate pathological condition is osteochondral destruction, but the process and therapeutic targets vary for each disease. With the arrival of biological drugs, arthritis treatment has drastically improved over the past 20 years. Novel treatments, such as drugs that target intracellular signals and nucleic acid drugs, have emerged. Their development toward clinical application is progressing at a rapid pace. It will be interesting to observe what the next paradigm shift in arthritis treatment will reveal.

## Figures and Tables

**Figure 1 cells-08-00818-f001:**
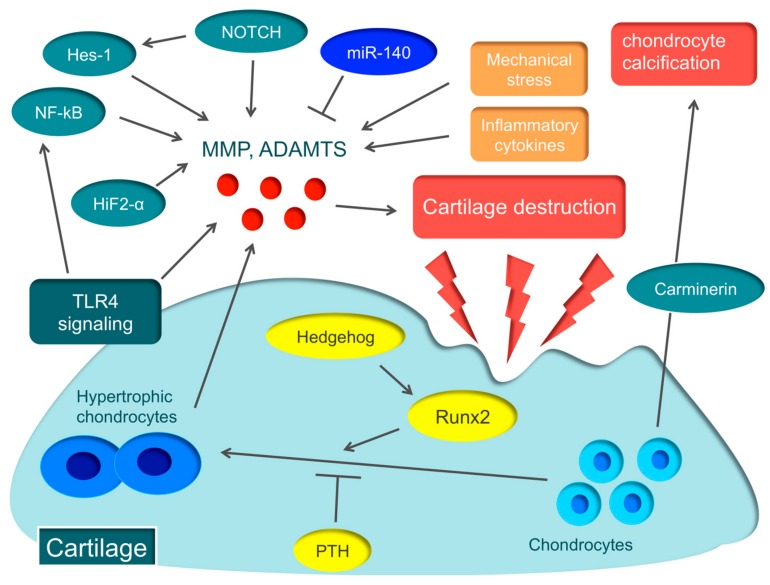
Various molecular mechanisms associated with osteoarthritis (OA) Hypertrophic chondrocytes are found in abundance in the early stage of articular cartilage destruction, from which matrix metalloproteinase (MMP) and a disintegrin and metalloproteinase with thrombospondin motifs (ADAMTS) are secreted, triggering OA. The essential transcription factor in the hypertrophy of chondrocytes is runt-related transcription factor 2 (Runx2) that is regulated by Hedgehog. On the other hand, parathyroid hormone (PTH) suppresses the hypertrophy of chondrocytes. Hypoxia-inducible factor-2α (HIF-2α), nuclear factor-kappa B (NF-κB), Notch, Hes1, and toll-like receptor 4 (TLR4) signaling promote the secretion of various degrading enzymes, including MMP and ADAMTS, while miR-140 protects cartilages by suppressing the expression of ADAMTS5. Carminerin is a cartilage-specific protein involved in chondrocyte calcification.

**Figure 2 cells-08-00818-f002:**
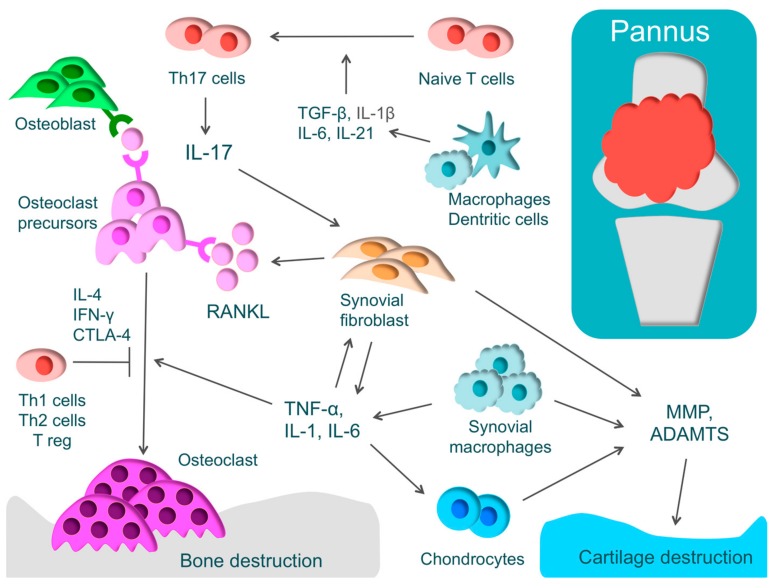
The pathology of rheumatoid arthritis (RA) and the mechanism of cartilage and bone destruction. RA is characterized by proliferative synovium (pannus) and an excessive immune response of T-cells. Pannus comprises T-cells, synovial fibroblasts, and macrophages that produce inflammatory cytokines, such as tumor necrosis factor (TNF)-α, interleukin-1 (IL-1), IL-6, and IL-17. These inflammatory cytokines activate osteoclasts, leading to bone destruction. T-helper (Th) cells have subsets such as Th1, Th2, and Th17 cells. Naive T-cells differentiate into Th17 cells through IL-1β, IL-6, IL-21, and transforming growth factor-β (TGF-β). Th17 cells produce IL-17 that activates inflammation by acting on various immune cells and activates osteoclasts by inducing the receptor activator of nuclear factor kappa B ligand (RANKL) in synovial fibroblasts. IFN-γ, IL-4, and cytotoxic T-lymphocyte-associated protein 4 (CTLA-4), produced by Th1, Th2, and Treg, respectively, regulate osteoclast differentiation. Cartilage destruction is caused by matrix metalloproteinase (MMP) and a disintegrin and metalloproteinase with thrombospondin motifs (ADAMTS) produced by chondrocytes, synovial fibroblasts, and synovial macrophages.

**Figure 3 cells-08-00818-f003:**
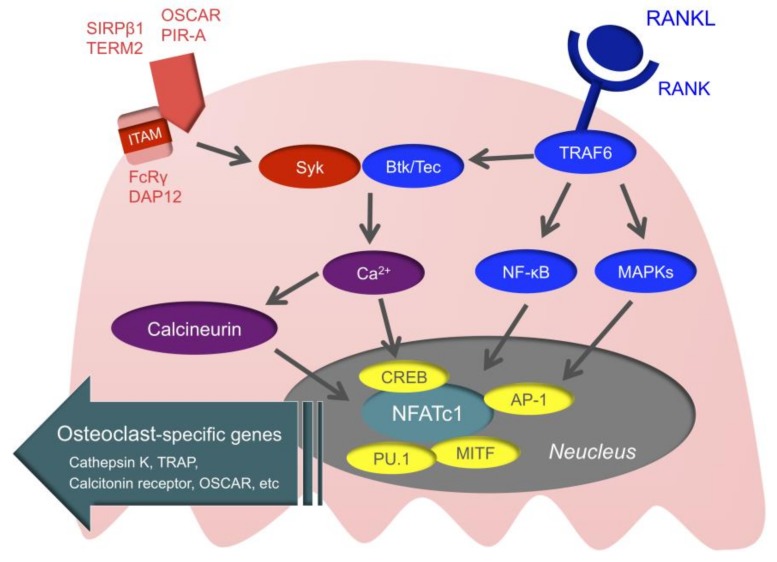
Intracellular signals of osteoclasts. When the receptor activator of nuclear factor kappa B (RANK) is stimulated, TNF receptor-associated factor 6 (TRAF6) is recruited to RANK. Then, mitogen-activated kinases (MAPKs), nuclear factor-kappa B (NF-κB), and activator protein1 (AP-1) are activated, and ultimately, the nuclear factor of activated T-Cell c1 (NFATc1) is activated. NFATc1 couples with AP-1, microphthalmia-associated transcription factor (MITF), PU.1, and cyclic adenosine monophosphate-response element-binding protein (CREB) and promotes the expression of osteoclast-specific genes. Adaptor molecules with immunoreceptor tyrosine-based activation motif (ITAM), such as DNAX-activating protein of 12 (DAP12) and Fc receptor common gamma chain (FcRγ), are associated with immunoglobulin-like receptors, such as osteoclast associated receptor (OSCAR), paired immunoglobulin-like receptor-A (PIR-A), triggering receptor expressed on myeloid cells 2 (TREM2), and signal regulatory protein beta 1 (SIRPβ1). Bruton’s tyrosine kinase (Btk) and tyrosine kinase expressed in hepatocellular carcinoma (Tec) integrate RANK/RANKL signaling with ITAM signaling.

**Figure 4 cells-08-00818-f004:**
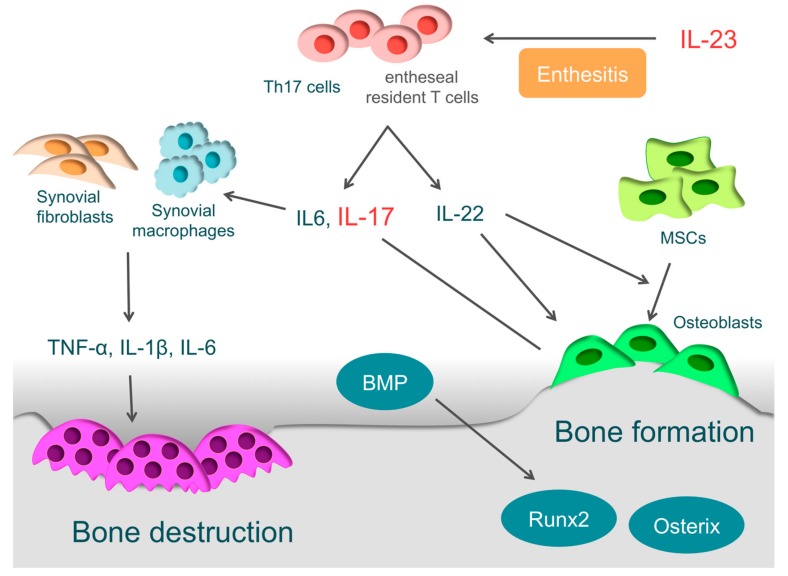
The pathology of psoriatic arthritis. In the pathology of psoriatic arthritis (PsA), T-helper (Th) 17 cells and the associated interleukin-23 (IL-23)/IL-17 axis are important. IL-23 is closely involved in the enthesitis. IL-23 drives entheseal resident T-cells expressing the IL-23 receptor to produce inflammatory cytokines, such as IL-6, IL-17, and IL-22. IL-17 drives synovial fibroblasts and macrophages to promote inflammatory cytokines, such as IL1-β, IL-6, and tumor necrosis factor (TNF)-α, and it ultimately causes bone destruction. IL-22 is involved in new bone formation in entheses or around articular cartilages. IL-22 promotes the proliferation of human mesenchymal stem cells (MSCs) and induces differentiation to osteoblasts. Furthermore, it activates osteoblasts via the signal transducer and activator of transcription 3. Bone morphogenetic protein (BMP), which strongly induces the expression of osteogenic transcription factors such as Runx2 and osterix, is also involved.

**Table 1 cells-08-00818-t001:** Summary of novel therapeutic agents.

Drug	Mechanism
*Novel Therapeutic Agents for OA*	
TNF-α inhibitors	Neutralize the biological activities of TNF-α by binding with soluble TNF-α and induce apoptosis, antibody-dependent cellular cytotoxicity and complement-dependent cytotoxicity in TNF-α-producing cells by binding to membrane-bound TNF-α
IL-1 inhibitors	Diacerein, Anakinra and AMG108 inhibit IL-1 pathways in inflammation respectively as an IL-1βinhibitor, an IL-1 receptor antagonist and an IL-1 receptor monoclonal antibody
Bisphosphonate	Suppresses the activity of osteoclasts, delay bone remodeling, and provides chondroprotective effects
Strontium ranelate	Induces cartilage formation through an ionic effect. Reduces osteoclastic bone resorption and simultaneously stimulate osteoblastic bone formation
Anti-NGF antibodies	Block nerve growth in intra-articular tissues and downregulate pain sensitivity
Therapies targeting TLR4 signaling	Inhibit TLR4 signaling by blocking TLR4 agonists, activating antagonist pathways and new inhibitory compounds
miR-140	Protect cartilages by suppressing the expression of ADAMTS5
Cell-based therapy	Chondrogenic potential and immunomodulatory properties of MSCs
*Novel Therapeutic Agents for RA*	
Btk inhibitors	Inhibit the differentiation and activation of osteoclasts by blocking the integration of RANK/RANKL signaling and ITAM signaling. Block the cytokine production and expression of co-stimulators via B cell receptors
Syk inhibitors	Block signal transduction for B-cell receptors, FcRγ, DAP12, and integrin with ITAM
PI3K inhibitors	Block activation of PI3K/Akt signaling pathway and suppress osteoclast formation and proliferation of B lymphocytes and synovial fibroblasts
miR-146	Functions as a negative feedback that stops the inflammatory stimulation caused by TNF-α by targeting TRAF6
miR-155	Suppresses the production of inflammatory cytokines by targeting SHIP1 and SOCS1
HDAC inhibitors	Promote the degradation of mRNA and regulate the generation of inflammatory cytokines in RA synovial fibroblasts and macrophage
*Novel Therapeutic Agents for PsA*	
JAK inhibitors	Block the JAK/STAT pathway that is the major signaling cascade for various pro-inflammatory cytokines
IL-6 inhibitors	Bind to both transmembrane IL-6 receptors and soluble IL-6 receptors to block IL-6 mediated signal transduction involving acute-phase response and activation of immune reaction
T cell activation inhibitors (Abatacept)	Inhibit the activity of T-cells that induce production of cytokines, autoantibodies, and inflammatory proteins
Tyk2 inhibitors	Block signaling downstream of the receptors for IL-12, IL-23, and type I and III interferons
CCL20 inhibitors	Bind to CCL20 and inhibit the movement of inflammatory cells into inflamed tissues
microbiota transplantation	Keeps gut homeostasis and regulate the activation of the inflammatory pathways

TNF (tumor necrosis factor), IL (interleukin), NGF (nerve growth factor), TLR (Toll-like receptor), ADMTS (a disintegrin and metalloproteinase with thrombospondin motifs), MSC (mesenchymal stem cell), Btk (Bruton’s tyrosine kinase), RANKL (receptor activator of nuclear factor kappa B ligand), ITAM (immunoreceptor tyrosine-based activation motif), Syk (spleen tyrosine kinase), FcRγ (Fc receptor common gamma subunit), DAP12 (DNAX-activating protein of 12), PI3K (phosphoinositide 3-kinase), TRAF (TNF receptor-associated factor), SHIP1 (src homology-2 domain-containing inositol 5-phosphatase 1), SOCS1 (suppressor of cytokine signaling 1), HDAC (histone deacetylase), JAK (janus kinase), STAT (signal transducer and the activator of transcription), Tyk (tyrosine kinase), CCL20 (chemokine CC motif ligand 20).

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
