# Peer review of "Cartilage and Bone Destruction in Arthritis: Pathogenesis and Treatment Strategy: A Literature Review"

_cells, 2019, doi:10.3390/cells8080818_

Round 1

Reviewer 1 Report

The article “Cartilage and Bone Destruction in Arthritis: 2 Pathogenesis and Treatment Strategy: A Literature Review” by Daisuke Tateiwa and colleagues was reviewed. The article provides a broad overview of major causes of osteoarthritis, rheumatoid arthritis and psoriatic arthritis, all revolving around the theme that inflammation in all cases leads to cartilage and subchondral bone destruction. The article is well written and would be of interest to readers of Cells interested in orthopaedics, cartilage biology, bone biology, inflammatory diseases and arthritis in general. Figures are appropriate and the sections on RA and PsA seem more or less complete. The manuscript should be considered for publication, however a few points must be addressed.

1.      The introduction should contain a very brief overview dedicated to cartilage biology and mechanics, also describing socio-economic impact of arthritis in general.

2.      In the first section on osteoarthritis, the authors should include a section on the role of toll-like receptors and matrix fragmentation in disease progression, and likewise therapeutics directed toward this end.

3.      Also, in the section of osteoarthritis, the authors failed to mention novel cell-based therapies, stem cell therapy and tissue engineering approaches with scaffolds. Furthermore, there is no summary on the gold standard surgical therapy for endstage arthritis – arthroplasty.

4.      To accommodate this, tables and figures may be modified as well to reflect this treatment strategies.

Author Response

<Reviewer1>

The article “Cartilage and Bone Destruction in Arthritis: 2 Pathogenesis and Treatment Strategy: A Literature Review” by Daisuke Tateiwa and colleagues was reviewed. The article provides a broad overview of major causes of osteoarthritis, rheumatoid arthritis and psoriatic arthritis, all revolving around the theme that inflammation in all cases leads to cartilage and subchondral bone destruction. The article is well written and would be of interest to readers of Cells interested in orthopaedics, cartilage biology, bone biology, inflammatory diseases and arthritis in general. Figures are appropriate and the sections on RA and PsA seem more or less complete. The manuscript should be considered for publication, however a few points must be addressed.

1.     The introduction should contain a very brief overview dedicated to cartilage biology and mechanics, also describing socio-economic impact of arthritis in general. 

Answer to comment 1:

Thank you for your valuable comment. According to your comment, we revised the manuscript as follows.

(Lines 44-45, Page 2, Introduction)

"Various molecular mechanisms behind cartilage destruction, including mechanical stress and inflammation, have been elucidated. " 

(Line 28-33, Page 1, Introduction)

"The morbidity rates of OA and RA are reported to be 10%–12% [4, 5]and 0.5%–1% of the adult population, respectively [6]. The socioeconomic burden of arthritis as well as the cost of pharmacological and surgical treatment (biological agents, joint arthroplasties, etc.) is enormous [7, 8]. Indeed, the cost of OA is estimated to account for 1%–2.5% of the gross national product in the United States, Canada, the United Kingdom, France, and Australia[4, 9]."

2.     In the first section on osteoarthritis, the authors should include a section on the role of toll-like receptors and matrix fragmentation in disease progression, and likewise therapeutics directed toward this end.

Answer to comment 2:

Thank you for your valuable comment. According to your comment, we revised the manuscript as follows.

(Lines 109-137, Page 2-3, 2.2.4. The role of toll-like receptors)

“2.2.4. The role of toll-like receptors (TLRs)

TLRs are type I transmembrane receptor proteins that play a key role in the innate immune response by recognizing pathogen-associated molecular patterns (PAMPs). In humans, the TLR family comprises 10 members (TLR1–TLR10), and some of them recognize host-derived damage-associated molecular patterns (DAMPs) that are produced when tissues are damaged [30]

Recently, the involvement of TLR4 in the development and progression of OA has been reported [30]. TLR4 was originally identified as a molecule that recognized lipopolysaccharide (LPS), which is a common component of the gram-negative bacterial cell wall. However, a distinct feature of TLR4 is that it binds to several PAMPs other than LPS and to various DAMPs, including high mobility group protein B1 (HMGB1), heat shock protein, free fatty acids, and extracellular matrix degradation products, such as fibronectin and hyaluronic acid [30]. When TLR4 binds to its agonists (PAMPs and DAMPs), it becomes activated. Subsequently, TLR4 signaling activates various transcription factors, including interferon regulatory factor (IRF), mitogen-activated protein kinase (MAPK), and NF-κB, promoting innate immune responses and inducing the production of inflammatory cytokines [30, 31]

TLR4 is highly expressed in immune cells, but it is also expressed in chondrocytes, synoviocytes, and osteoblasts [32]. It has been shown that TLR4 is highly expressed in cartilage lesions in OA patients [33]. In fact, activation of TLR4 signaling in human chondrocytes promotes the production of inflammatory cytokines and inhibits the synthesis of aggrecan and type II collagen [33]. Furthermore, it has been reported that there is an association between mechanical stress and TLR4 signaling, and that high shear stress promotes TLR4-dependent production of IL-6 in human chondrocytes [32]

In cartilage, several DAMPs activate TLR4 signaling, thereby producing inflammatory cytokines, causing cartilage degradation. Small hyaluronic acid fragments can activate TLR4 in chondrocytes, which induces an inflammatory response. However, high molecular weight hyaluronic acid exerts an inhibitory effect on the inflammatory response, suggesting that the action of hyaluronic acid on TLR4 signaling may differ depending on its molecular size [34]. Other TLR4 agonists involved in OA conditions include fibronectin fragments, HMGB1, serum amyloid A, S100-A8, S100-A9, and some plasma proteins contained in synovial fluid [30].

(Line 269-278, Page 7, 2.5.5 Therapies targeting TLR signaling).

“2.5.5. Therapies targeting TLR signaling

TLR signaling, in particular TLR4-mediated signaling, is involved in the development and progression of OA. Recently, the efficacy of treatments targeting TLR4 signaling has been reported in preclinical studies [76-78]. Pep-1 (hyaluronic acid-binding peptide) inhibits the degradation of hyaluronic acid and consequently reduces the small hyaluronic acid fragmentation ratio [76]. PPAR-γagonists (genistein and rosiglitazone) inhibit TLR4 signaling by reducing serum amyloid A expression [77]. Vasoactive intestinal polypeptide (VIP) decreases TNF-α-induced TLR4 expression in synoviocytes from OA patients [78]. Although the efficacy of new compounds that regulate TLR4 signaling, such as oleochanthal, 6-shogaol, boswellic acid, quercetin, and kampferol, has been reported, the mechanisms have not yet been fully elucidated [30, 79, 80].

3.     Also, in the section of osteoarthritis, the authors failed to mention novel cell-based therapies, stem cell therapy and tissue engineering approaches with scaffolds. Furthermore, there is no summary on the gold standard surgical therapy for end stage arthritis – arthroplasty.

Answer to comment 3:

Thank you for your valuable comment. According to your comment, we revised the manuscript as follows.

(Line 303-328, Page 7-8, 2.5.7. Cell-based therapy)

“2.5.7. Cell-based therapies

Cell-based therapies for OA have advanced remarkably in recent years. Currently, the only cell-based therapy approved by the Food and Drug Administration (FDA) is autologous chondrocyte implantation (ACI), whereby cultured chondrocytes are applied to the injured cartilage area [90]. Although long-term results are lacking, several clinical trials have reported the short-term efficacy of ACI for OA treatment [91].

In recent years, mesenchymal stem cells (MSCs) have attracted the most attention in cell-based therapies for OA [92, 93]. MSCs are well known as multipotent cells with self-replication potential. Furthermore, MSCs possess immunomodulatory properties that suppress the immune response and reduce the production of inflammatory cytokines. Although MSCs can be isolated from various tissues, bone marrow-derived MSCs (BM-MSCs) and adipose-derived MSCs (AD-MSCs) are most commonly used [94]. In a comparison between BM-MSCs and AD-MSCs, BM-MSCs showed superior chondrogenic and osteogenic ability, while AD-MSCs were superior in terms of immunomodulatory properties [95].

There are two main methods to deliver MSCs into the joints: intra-articular injection of MSCs in suspension and surgical procedures that implant MSC-laden scaffolds [93]. Intra-articular injection of MSCs is relatively simple, less invasive, and more cost effective compared with surgical procedures [93]. On the contrary, in surgical procedures using scaffolds, the properties of the scaffolds can result in excellent therapeutic effects. An ideal scaffold should initially promote cell migration and provide biological, chemical, and mechanical support to MSCs. It should also be finally absorbed or biodegraded so as not to hamper new cartilage formation or induce an inflammatory response [96].

 According to recent preclinical and clinical trials, OA treatment using MSCs has demonstrated a significant improvement in clinical results, with minimal adverse events [92, 93, 97, 98]. However, more reliable studies with randomized controls and larger sample sizes are required for robust evidence.

(Lines 208-213, Page 5, 2.4. Present therapeutic strategy for OA).

“Surgical management is considered for OA patients who do not attain adequate pain relief and functional improvement with a combination of non-pharmacological and pharmacological treatment [49, 54]. Joint replacement surgery is available for patients with severe symptoms or functional limitations associated with poor health-related quality of life despite conservative treatment. Osteotomy and joint preserving surgeries are considered for young and physically active patients with symptomatic OA [54].”

4.     To accommodate this, tables and figures may be modified as well to reflect this treatment strategies.

Answer to comment 4:

Thank you for your valuable comment. According to your comment, we modified Table 1 and Figure 1.

(Page 18, Table 1)

Table 1: We added “Therapies targetingTLR4 signaling” and “Cell-based therapy”.

(Page 5, Figure 1)

Figure 1: We added “TLR signaling”.

Reviewer 2 Report

The review is exhaustive with very up to date bibliographic references on the subject, but in terms of contribution to the knowledge does not provide a great scientific advance.

Once again, rheumatic inflammation treatments are very generic and are not exempt from secondary risks. It does not escape to me that this is a review of the current state of the art, but it would have been more interesting to put the focus on new therapeutic, such as cell therapies and other pathways, such as Wnt related pathways. A comprehensive review of new experimental strategies will be more valuable for the readers. 

---------------------------------------------------------------------------------------

The abstract and the first part of the introduction are very similar (Cut&Paste action)

The typo thrombospindin must be changed by thrombospondin

Page 1, line 14

Page 1, line 32

Author Response

<Reviewer2>

The review is exhaustive with very up to date bibliographic references on the subject, but in terms of contribution to the knowledge does not provide a great scientific advance.

Once again, rheumatic inflammation treatments are very generic and are not exempt from secondary risks. It does not escape to me that this is a review of the current state of the art, but it would have been more interesting to put the focus on new therapeutic, such as cell therapies and other pathways, such as Wnt related pathways. A comprehensive review of new experimental strategies will be more valuable for the readers. 

Answer:

Thank you for your valuable comment. According to your comment, we have added more information about new cell therapies and other pathways (toll-like receptor signaling pathway). These sections contain experimental strategies. We revised the manuscript as follows.

(Lines 303-328, Page 7-8, 2.5.7. Cell-based therapy)

“2.5.7. Cell-based therapies

Cell-based therapies for OA have advanced remarkably in recent years. Currently, the only cell-based therapy approved by the Food and Drug Administration (FDA) is autologous chondrocyte implantation (ACI), whereby cultured chondrocytes are applied to the injured cartilage area [90]. Although long-term results are lacking, several clinical trials have reported the short-term efficacy of ACI for OA treatment [91].

In recent years, mesenchymal stem cells (MSCs) have attracted the most attention in cell-based therapies for OA [92, 93]. MSCs are well known as multipotent cells with self-replication potential. Furthermore, MSCs possess immunomodulatory properties that suppress the immune response and reduce the production of inflammatory cytokines. Although MSCs can be isolated from various tissues, bone marrow-derived MSCs (BM-MSCs) and adipose-derived MSCs (AD-MSCs) are most commonly used [94]. In a comparison between BM-MSCs and AD-MSCs, BM-MSCs showed superior chondrogenic and osteogenic ability, while AD-MSCs were superior in terms of immunomodulatory properties [95].

There are two main methods to deliver MSCs into the joints: intra-articular injection of MSCs in suspension and surgical procedures that implant MSC-laden scaffolds [93]. Intra-articular injection of MSCs is relatively simple, less invasive, and more cost effective compared with surgical procedures [93]. On the contrary, in surgical procedures using scaffolds, the properties of the scaffolds can result in excellent therapeutic effects. An ideal scaffold should initially promote cell migration and provide biological, chemical, and mechanical support to MSCs. It should also be finally absorbed or biodegraded so as not to hamper new cartilage formation or induce an inflammatory response [96].

 According to recent preclinical and clinical trials, OA treatment using MSCs has demonstrated a significant improvement in clinical results, with minimal adverse events [92, 93, 97, 98]. However, more reliable studies with randomized controls and larger sample sizes are required for robust evidence.”

(Line 109-137, Page 3-4, 2.2.4. The role of toll-like receptors)

“2.2.4. The role of toll-like receptors (TLRs)

TLRs are type I transmembrane receptor proteins that play a key role in the innate immune response by recognizing pathogen-associated molecular patterns (PAMPs). In humans, the TLR family comprises 10 members (TLR1–TLR10), and some of them recognize host-derived damage-associated molecular patterns (DAMPs) that are produced when tissues are damaged [30]

Recently, the involvement of TLR4 in the development and progression of OA has been reported [30]. TLR4 was originally identified as a molecule that recognized lipopolysaccharide (LPS), which is a common component of the gram-negative bacterial cell wall. However, a distinct feature of TLR4 is that it binds to several PAMPs other than LPS and to various DAMPs, including high mobility group protein B1 (HMGB1), heat shock protein, free fatty acids, and extracellular matrix degradation products, such as fibronectin and hyaluronic acid [30]. When TLR4 binds to its agonists (PAMPs and DAMPs), it becomes activated. Subsequently, TLR4 signaling activates various transcription factors, including interferon regulatory factor (IRF), mitogen-activated protein kinase (MAPK), and NF-κB, promoting innate immune responses and inducing the production of inflammatory cytokines [30, 31]

TLR4 is highly expressed in immune cells, but it is also expressed in chondrocytes, synoviocytes, and osteoblasts [32]. It has been shown that TLR4 is highly expressed in cartilage lesions in OA patients [33]. In fact, activation of TLR4 signaling in human chondrocytes promotes the production of inflammatory cytokines and inhibits the synthesis of aggrecan and type II collagen [33]. Furthermore, it has been reported that there is an association between mechanical stress and TLR4 signaling, and that high shear stress promotes TLR4-dependent production of IL-6 in human chondrocytes [32]

In cartilage, several DAMPs activate TLR4 signaling, thereby producing inflammatory cytokines, causing cartilage degradation. Small hyaluronic acid fragments can activate TLR4 in chondrocytes, which induces an inflammatory response. However, high molecular weight hyaluronic acid exerts an inhibitory effect on the inflammatory response, suggesting that the action of hyaluronic acid on TLR4 signaling may differ depending on its molecular size [34]. Other TLR4 agonists involved in OA conditions include fibronectin fragments, HMGB1, serum amyloid A, S100-A8, S100-A9, and some plasma proteins contained in synovial fluid [30].”

(Line 269-278, Page 7, 2.5.5. Therapies targeting TLR signaling).

“2.5.5. Therapies targeting TLR signaling

TLR signaling, in particular TLR4-mediated signaling, is involved in the development and progression of OA. Recently, the efficacy of treatments targeting TLR4 signaling has been reported in preclinical studies [76-78]. Pep-1 (hyaluronic acid-binding peptide) inhibits the degradation of hyaluronic acid and consequently reduces the small hyaluronic acid fragmentation ratio [76]. PPAR-γagonists (genistein and rosiglitazone) inhibit TLR4 signaling by reducing serum amyloid A expression [77]. Vasoactive intestinal polypeptide (VIP) decreases TNF-α-induced TLR4 expression in synoviocytes from OA patients [78]. Although the efficacy of new compounds that regulate TLR4 signaling, such as oleochanthal, 6-shogaol, boswellic acid, quercetin, and kampferol, has been reported, the mechanisms have not yet been fully elucidated [30, 79, 80].

---------------------------------------------------------------------------------------

The abstract and the first part of the introduction are very similar (Cut&Paste action)

Answer:

Thank you for your valuable comment. According to your comment, we have modified the abstract and the first part of the introduction.

The typo thrombospindin must be changed by thrombospondin

Page 1, line 14

Page 1, line 32

Answer:

Thank you for your valuable comment. According to your comment, we revised “thrombospindin” to “thrombospondin”.

Reviewer 3 Report

Cells-549558

Cartilage and Bone Destruction in Arthritis: Pathogenesis and Treatment Strategy: A Literature Review

This review article summarizes known facts concerning pathogenesis, molecular mechanisms, and treatment of osteoarthritis (OA), rheumatoid arthritis (RA), and psoriatic arthritis (PsA) with a focus on cartilage and bone destruction. Pros and cons of current and possible future therapeutic approaches are also discussed.

Text and figures appear straightforward and clear. The review article covers the selected topic, reflects relevant parts of the latest literature, and provides an informative overview for the reader. Therefore, only a few a few aspects should be addressed.

1. In some parts, the text contains redundant information that could be shortened (e.g., the mention of inflammation-associated osteochondral destruction on page 1, lines 9, 11, and 29; in the phrase “… secreted MMPs that are secreted outside cells …” on page 2, line 80; etc.).

2. The figures should be linked to the text. In Figure 1, the connection between Notch and Hes-1 and in Figure 2, the cell-to-cell contacts between osteoblasts and osteoclast progenitors (mentioned on page 7, line 272) should be included.

3. For OA and RA, short paragraphs discussing the influence of the genetic background should be included.

4. In subsection 2.3 (Molecular mechanisms associated with OA), it should be specified which NF-kappaB subunits are involved and whether there is a difference between canonical and non-canonical NF-kappaB signalling.

5. In subsection 3.2 (Mechanisms of bone destruction) and in Figure 2, MMP production by synovial fibroblasts should be mentioned.

6. In subsection 3.4.5 (T-cell activation inhibitors), please specify the mode of action by which abatacept is regulating T-cell activity.

7. Concerning subsections 3.5.3 (PI3K inhibitors) and 3.5.5 (HDAC inhibitors): are there clinical studies testing specific PI3K and HDAC inhibitors?

8. In subsection 3.5.4 (miRNA), the relevant molecular targets of miR-155 should be provided.

9. In subsection 4.7.1 (Using appropriate biological agents), please specify the “..different biological agents …” mentioned in line 594/595.

10. Several pharmaceuticals (especially inhibitors) are only mentioned in general (e.g., PI3K, HDAC, or PDE4 inhibitors). Please provide some examples.

11. In general, known adverse effects of the therapeutic approaches (if any) should be mentioned or discussed in more detail.

Author Response

<Reviewer3>

Cartilage and Bone Destruction in Arthritis: Pathogenesis and Treatment Strategy: A Literature Review

This review article summarizes known facts concerning pathogenesis, molecular mechanisms, and treatment of osteoarthritis (OA), rheumatoid arthritis (RA), and psoriatic arthritis (PsA) with a focus on cartilage and bone destruction. Pros and cons of current and possible future therapeutic approaches are also discussed.

Text and figures appear straightforward and clear. The review article covers the selected topic, reflects relevant parts of the latest literature, and provides an informative overview for the reader. Therefore, only a few a few aspects should be addressed.

1.     In some parts, the text contains redundant information that could be shortened (e.g., the mention of inflammation-associated osteochondral destruction on page 1, lines 9, 11, and 29; in the phrase “… secreted MMPs that are secreted outside cells …” on page 2, line 80; etc.).

Answer to comment 1:

Thank you for your valuable comment. According to your comment, we have reviewed the whole text again and omitted the redundant information and phrases as follows.

(Line 14, Page 1, Abstract)

 “…various inflammatory cytokines, such as tumor necrosis factor, interleukins, matrix metalloproteinase, and a disintegrin and metalloproteinase with thrombospondin motifs.”

(Line 93-94, Page 3, 2.2.2. ECM degradation by MMP),

“… secreted MMPs that are secreted outside cells…”, “…membrane-associated MMPs that are bound to the cell membrane…”, etc.

2.     The figures should be linked to the text. In Figure 1, the connection between Notch and Hes-1 and in Figure 2, the cell-to-cell contacts between osteoblasts and osteoclast progenitors (mentioned on page 7, line 272) should be included.

Answer to comment 2:

Thank you for your valuable comment. According to your comment, we have revised the figures. We specified the connection between Notch and Hes-1 (Figure 1) and the cell-to-cell contacts between osteoblasts and osteoclast progenitors (Figure 2).

3.     For OA and RA, short paragraphs discussing the influence of the genetic background should be included.

Answer to comment 3:

Thank you for your valuable comment. According to your comment, we revised the manuscript as follows.

(Line 64-71, Page 2, 2.1.1 Genetic factors in OA)

“2.1.1. Genetic factors in OA

The involvement of genetic factors in OA has been reported. A classic twin study described an apparent genetic effect for OA, with a genetic influence ranging from 39%–65% after adjusting for known risk factors [18]. To date, Genome Wide Association Studies (GWAS) have identified several genetic variants associated with OA, although it has been observed that the identified individual risk alleles only exerted moderate to slight effects on overall OA susceptibility [19]. This information can be applied to detect individuals at high risk of developing OA and subsequently implement preventive or interventional treatments [17].”

(Page 8, Line 351-357, 3.1.1 Genetic factors in RA)

3.1.1. Genetic factors in RA

Genetic factors are closely involved in the onset of RA. It is estimated that genetic factors contribute to 50% of the risk of developing RA [102]. A nationwide twin study reported that disease concordance in monozygotic twins (15.4%) is considerably higher than in dizygotic twins (3.6%)[103]. Among MHC class II genes, HLA-DR1 and HLA-DR4 are most closely connected to RA susceptibility [104]. To date, in multiple populations, the GWAS have identified more than 35 RA susceptibility loci, such as HLA-DRB1STAT4PTPN22,PAD14, and CCR6[105, 106].

4.     In subsection 2.3 (Molecular mechanisms associated with OA), it should be specified which NF-kappaB subunits are involved and whether there is a difference between canonical and non-canonical NF-kappaB signalling.

Answer to comment 4:

Thank you for your valuable comment. According to your comment, we revised the manuscript as follows.

(Line 158-166, Page 4, 2.3. Molecular mechanisms associated with OA)

“NF-κB is a ubiquitously expressed transcription factor that regulates inflammation, immune responses, cell proliferation, and survival [44]. The NF-κB family comprises five subunits: RelA/p65, RelB, c-Rel, p50/p105 (NF-κB1), and p52/p100 (NF-κB2). There are two distinct pathways that activate NF-κB, the canonical (classical) pathway and the non-canonical (alternative) pathway. The canonical pathway is activated by inflammatory cytokinesand promotes the secretion of various degrading enzymes, including MMP and ADAMTS, while suppressing the expression of ECM-synthesizing molecules, such as Sox9, and the synthesis of type II collagen and aggrecan [45]. In fact, in a murine OA model, knockdown of NF-kBp65with siRNA suppresses cartilage destruction [40].”

5.     In subsection 3.2 (Mechanisms of bone destruction) and in Figure 2, MMP production by synovial fibroblasts should be mentioned.

Answer to comment 5:

Thank you for your valuable comment. According to your comment, we revised Figure 2 and the manuscript as follows.

(Line 386-388, Page 9, 3.2. Mechanism of bone and cartilage destruction)

“Cartilage destruction is caused by MMP or ADAMTS, which are produced by chondrocytes, synovial fibroblasts, and synovial macrophages. Epigenetic changes that maintain high levels of MMP expression have been found in RA synovial fibroblasts [116].”

6.     In subsection 3.4.5 (T-cell activation inhibitors), please specify the mode of action by which abatacept is regulating T-cell activity.

Answer to comment 6:

Thank you for your valuable comment. According to your comment, we revised the manuscript as follows.

(Line 503-508, Page 12, 3.4.4. T-cell activation inhibitors)

“Co-stimulatory signals are essential for T-cell activation, and without co-stimulatory signals, T-cells undergo apoptosis. CD28, which is expressed in T-cells, is the archetypal co-stimulatory molecule that binds to CD80 and CD86 on the surface of antigen-presenting cells. CTLA4 contained in abatacept binds to CD80 and CD86 with higher affinity than CD28, thereby acting as a negative regulator of the CD80/CD86:CD28 co-stimulatory signal [146].”

7.     Concerning subsections 3.5.3 (PI3K inhibitors) and 3.5.5 (HDAC inhibitors): are there clinical studies testing specific PI3K and HDAC inhibitors?

Answer to comment 7:

Thank you for your question. Clinical trials of PI3K inhibitors for RA have not been conducted yet. Furthermore, no clinical trials of HDAC inhibitors for RA have been conducted yet, but the efficacy and safety of givinostat (pan-HDAC inhibitor) have been reported in clinical trials for juvenile idiopathic arthritis. We revised the manuscript as follows.

(Line 550, Page 13, 3.5.3. Phosphoinositide 3-kinase(PI3K) inhibitors)

“Clinical trials of PI3K inhibitors for RA have not yet been conducted.”

(Line 568-571, Page 13, 3.5.5. Histone deacetylase (HDAC) inhibitors)

“Clinical trials of HDAC inhibitors of RA have not been conducted, but the efficacy and safety of givinostat have been reported in clinical trials for juvenile idiopathic arthritis[169]. In the trial,givinostat was safe and well tolerated. The most common adverse events were short-lived, self-limited respiratory or gastrointestinal disturbances.”

8.     In subsection 3.5.4 (miRNA), the relevant molecular targets of miR-155 should be provided.

Answer to comment 8:

Thank you for your valuable comment. According to your comment, we revised the manuscript as follows.

(Line 559-562, Page 13, 3.5.4. MicroRNA (miRNA))

“Overexpression of miR-155 in RA synovial monocytes and macrophages enhances the production of TNF-α, IL-6, IL-1β, and IL-8. On the other hand, inhibition of miR-155 suppresses the production of TNF-α [164]. Src homology-2 domain-containing inositol 5-phosphatase 1 (SHIP1) and suppressor of cytokine signaling 1 (SOCS1) have been identified as major targets of miR-155 [165, 166].”

9.     In subsection 4.7.1 (Using appropriate biological agents), please specify the “..different biological agents …” mentioned in line 594/595.

Answer to comment 9:

Thank you for your valuable comment. According to your comment, we revised the manuscript as follows.

(Line 703-704, Page 17, 4.6.1. Using appropriate biological agents)

“New evidence on the appropriate use of adalimumab and secukinumabis much anticipated.”

10.  Several pharmaceuticals (especially inhibitors) are only mentioned in general (e.g., PI3K, HDAC, or PDE4 inhibitors). Please provide some examples.

Answer to comment 10:

Thank you for your valuable comment. According to your comment, we revised the manuscript as follows.

(Line 531-532,Page 13, 3.5.1. Btk inhibitors)

“Currently, Btk inhibitors, such as poseltinib (HM71224), are in clinical development [152].”

(Line 537-538,Page 13, 3.5.2. Syk inhibitors)

“Although the efficacy of Syk inhibitors, such as fostamatinib disodium (R406),on RA has been reported [155]…”

(Line 546-547,Page 13, 3.5.3. Phosphoinositide 3-kinase(PI3K) inhibitors)

“The efficacy of PI3K inhibitors, such as ZSTK474,has been demonstrated in mice with collagen-induced arthritis [159].”

(Line 566-568,Page 13, 3.5.5. Histone deacetylase (HDAC) inhibitors)

“HDAC inhibitors, such as trichostatin A and givinostat (ITF2357),promote the degradation of mRNA and regulate the generation of inflammatory cytokines in RA synovial fibroblasts and macrophages [168].”

11.  In general, known adverse effects of the therapeutic approaches (if any) should be mentioned or discussed in more detail.

Answer to comment 11:

Thank you for your valuable comment. According to your comment, we revised the manuscript as follows.

(Line 254-256, Page 6, 2.5.3. Strontium ranelate)

“However, the increased risk of cardiac events, such as venous thrombosis, pulmonary embolism, and myocardial infarction, has been reported in randomized controlled trials, but not in real life; therefore, further studies are warranted [69].”

(Line 265-267, Page 7, 2.5.4. Anti-nerve growth factor (NGF) antibody)

“Tanezumab and fasinumab were generally well tolerated. The most common adverse events in tanezumab were headache, upper respiratory tract infections, and sensory abnormalities [73].”

(Line 537-539, Page 13, 3.5.2. Syk inhibitors)

“Although the efficacy of Syk inhibitors, such as fostamatinib disodium (R406), on RA has been reported [155],it has not been used due to its side effects, such as infections, hypertension, diarrhea, nausea, and headache [156].

(Page 13, Line 570-571,3.5.5. Histone deacetylase (HDAC) inhibitors)

“In the trial,givinostat was safe and well tolerated. The most common adverse events were short-lived, self-limited respiratory or gastrointestinal disturbances.”

(Line 728-730, Page 17, 4.6.2. Novel therapeutic agents)

“BMS-986165, a selective oral inhibitor of Tyk2, had a good therapeutic result in psoriasis that was the same as the results for other biological agents, such as ustekinumab and adalimumab, although an increased risk of mild-to-moderate acne has been reported in the treatment groups [222].”

Round 2

Reviewer 1 Report

no further comments